# Solitary Fibrous Tumor of the Prostate: A Case Report and Literature Review

**DOI:** 10.3390/medicina57111152

**Published:** 2021-10-23

**Authors:** Yasumichi Takeuchi, Daiki Kato, Keita Nakane, Kota Kawase, Manabu Takai, Koji Iinuma, Chiemi Saigo, Tatsuhiko Miyazaki, Takuya Koie

**Affiliations:** 1Department of Urology, Gifu University Graduate School of Medicine, Gifu 5011194, Japan; yastake@gifu-u.ac.jp (Y.T.); andreas7@gifu-u.ac.jp (D.K.); keitaco@gifu-u.ac.jp (K.N.); stnf55@gifu-u.ac.jp (K.K.); takai_mb@gifu-u.ac.jp (M.T.); kiinuma@gifu-u.ac.jp (K.I.); 2Department of Pathology and Translational Research, Gifu University Graduate School of Medicine, Gifu 5011194, Japan; saigou@gifu-u.ac.jp; 3Department of Pathology, Gifu University Graduate School of Medicine, Gifu 5011194, Japan; tats_m@gifu-u.ac.jp

**Keywords:** prostatic tumor, solitary fibrous tumor, rare case, case report

## Abstract

Solitary fibrous tumors (SFTs) usually occur in the pleura and account for two-thirds of all cases; however, SFTs occurring in the prostate are extremely rare. Approximately 25 cases have been reported in the literature to date. This study reports the case of a 43-year-old man referred to our hospital with the chief complaint of a pelvic tumor after careful examination. The tumor marker levels were within normal limits. T2-weighted magnetic resonance imaging revealed a tumor, demonstrating primarily low signal intensity. It showed a capsule-like rim at the left lobe of the prostate, suggesting that the tumor was partially invading the rectal wall. Histopathological examination of needle-core biopsies showed spindle cell neoplasm with small and fusiform cells, strongly expressing signal transducer and activator of transcription 6 (STAT6) with a ramifying vascular network. Therefore, the clinical diagnosis of the patient was SFT of the prostate and robot-assisted radical prostatectomy was performed. Histopathological examination revealed that the tumor was composed of spindle cells with patternless and staghorn patterns. Immunohistochemical analysis showed a strong expression of STAT6. Furthermore, the tumor was partially positive for CD34. Therefore, the patient was diagnosed with SFT of the prostate. Two years after the initial diagnosis, the patient was alive with normal erectile function, continence status, and no evidence of the disease.

## 1. Introduction

A solitary fibrous tumor (SFT) is a rare mesenchymal neoplasm of myofibroblastic origin [1] and was first reported in 1931 as a neoplasm originating from the pleura [2]. Although SFTs usually occur in the pleura and account for two-thirds of all cases, over the past decade, nearly 30% of SFTs have been reported to occur in the nonserosal and extrathoracic regions [3]. STF, a soft tissue lesion of mesenchymal origin, has been recently determined to be associated with a paracentric inversion involving chromosome 12q that results in a NAB2-STAT6 fusion gene [4,5]. Out of these, prostate SFTs are extremely rare, and approximately 25 such cases have been reported in the literature to date [6]. However, the diagnosis of mesenchymal neoplasms of the prostate is often difficult for pathologists because of their overlapping histomorphological and immunohistochemical (IHC) characteristics [7]. In fact, we previously encountered a patient with prostate SFT that was initially misdiagnosed as prostate cancer, based on histopathological examination of prostate biopsy specimen [8].

Here, we report a case of a 43-year-old man diagnosed with a prostatic tumor during a medical checkup and later with prostate SFT.

## 2. Case Report

A 43-year-old man was referred to our hospital with a chief complaint of pelvic tumor after careful examination. The tumor was identified during a medical checkup; however, the patient had no symptoms of urinary tract obstruction. On digital rectal examination, a large, soft, and movable mass without tenderness was palpable in the left lobe of the prostate. Tumor markers, including prostate-specific antigen (0.675 ng/mL; normal range <4.0 ng/mL), carcinoembryonic antigen, and carbohydrate antigen 19-9 levels, were found to be within normal limits.

T2-weighted magnetic resonance imaging revealed a tumor of 30 × 34 mm in size, demonstrating primarily low signal intensity, with a capsule-like rim at the left lobe of the prostate, suggesting that the tumor was partially invading the rectal wall (Figure 1).

Histopathological examination of ultrasound-guided needle core biopsies showed a spindle cell neoplasm with small and fusiform cells, strongly expressing signal transducers and activators of transcription 6 (STAT6) with a ramifying vascular network. Radiographic examination of the whole body revealed no lymphadenopathy or distant metastases. Therefore, the clinical diagnosis of the patient was SFT in the prostate.

Robot-assisted radical prostatectomy was performed to predict the risk of recurrence or metastasis based on clinically aggressive features [9,10]. Anterior resection of the rectum was not necessary, even though there was firm adhesion between the tumor and rectal wall (Figure 2). In this case, we could spare the neurovascular bundle in the right side. There were no surgery-related complications, including neurologic complications [11].

Macroscopic examination revealed a solid, yellowish-white tumor measuring 3 × 3.4 cm on the left side of the resected prostate (Figure 3).

Histopathological examination revealed a well-circumscribed mass composed of spindle cells arranged without any particular architecture (patternless), associated with variable amounts of collagen and distinctive vascularization (staghorn pattern) (Figure 4A,B).

Immunohistochemical (IHC) analysis showed a strong expression of STAT6 in the tumor cells, which were partially positive for CD34 (Figure 5A,B). Therefore, the patient was finally diagnosed with SFT in the prostate based on the histological and IHC findings.

The patient progressed favorably, and there were no obvious complications after surgery. Two years after the initial diagnosis, the patient was alive with normal erectile function, continence status, and no evidence of the disease.

## 3. Discussion

Mesenchymal neoplasms comprise <1% of tumors occurring in the prostate [12]. Although stromal proliferations or neoplasms are representative of the most common mesenchymal tumors in the prostate, SFT, myoblastic proliferation, smooth muscle neoplasms, gastrointestinal stromal tumors, schwannomas, rhabdomyosarcomas, and mixed epithelial stromal tumors of the seminal vesicle may be encountered in biopsy specimens that are obtained for needle biopsies from the prostate [12,13]. Microscopic and immunohistochemical characteristics of prostatic spindle-cell lesions are shown in Table 1 and Table 2 [12,13,14].

In particular, the distinction between the SFTs and prostatic stromal tumors of uncertain malignant potential may be problematic and will be facilitated by additional specific IHC markers [14].

Recent genomic studies on SFTs have identified a novel fusion transcript, NAB2-STAT6 (NGFI-A binding protein 2-nuclear signal transducer and STAT6) as an SFT-specific chimeric fusion gene [5,15]. Currently, a novel monoclonal antibody for STAT6 is available [16]. Therefore, strong STAT6 expression observed using IHC has been shown to be a highly sensitive and specific diagnostic marker for SFTs [5,16,17].

Although most SFTs usually follow a benign clinical course, a small number of SFTs may transform into more aggressive forms and have more adverse clinical outcomes [9,10,17,18,19]. Recently, risk stratification models have identified malignant SFT, which behaves aggressively [19]. The predictive factors for SFTs with malignant features include tumor size, anatomical location, mitotic activity, nuclear pleomorphism, necrosis, and patient age [12,19,20]. Demicco et al. [19] developed a risk stratification model based on patient age, tumor size, and mitotic figures. Patients older than 55 years with tumor size greater than 15 cm and mitotic counts greater than 4 mitoses/10 HPFs had the highest risk of metastases or tumor death. In addition, telomerase reverse transcriptase gene promoter mutations were strongly associated with higher age, larger tumor size, and higher risk classification [21]. Moereau-Zabotto et al. [22] have reported that out of 17 patients with SFTs who had follow-up after surgery, one (5.8%) patient experienced a local recurrence, and two (11.8%) patients died of postoperative complications. Therefore, complete resection of SFTs can be an important predictive factor for achieving better clinical outcomes [23].

## 4. Conclusions

We observed that SFT in the prostate could be removed with complete resection-negative surgical margins. However, 10% of SFTs with histological features of malignancy may be unpredictable, exhibiting aggressive behavior with local recurrence or distant metastasis. Therefore, a long-term follow-up is recommended for all patients with SFTs, including our case.

## Figures and Tables

**Figure 1 medicina-57-01152-f001:**
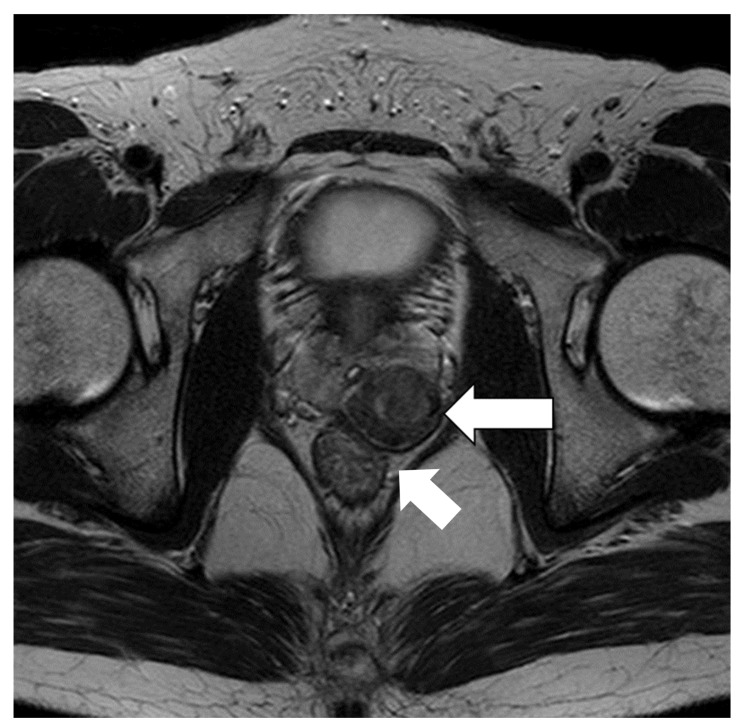
T2-weighted magnetic resonance image demonstrating the tumor mainly with low signal intensity (arrow). The tumor shows a capsule-like rim at the left lobe of the prostate, suggesting that it is partially invading the rectal wall (arrow).

**Figure 2 medicina-57-01152-f002:**
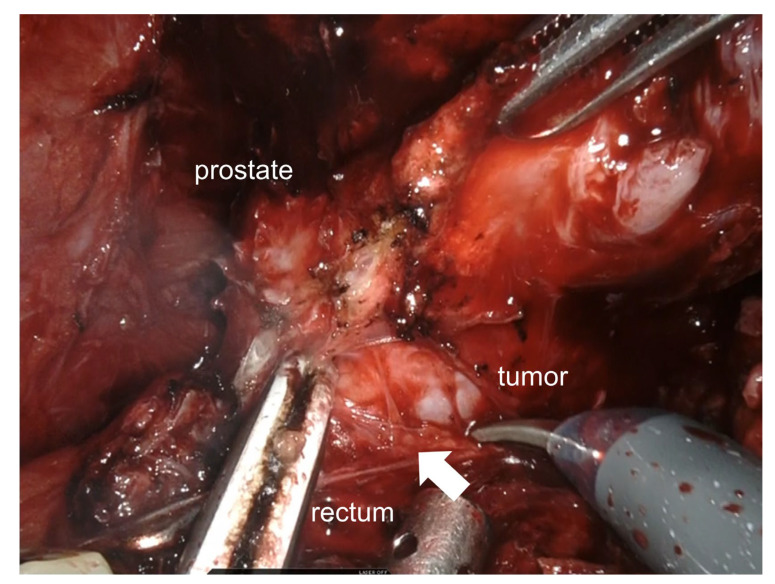
Robot-assisted radical prostatectomy findings. A firm adhesion between the tumor and the rectal wall is observed (arrow).

**Figure 3 medicina-57-01152-f003:**
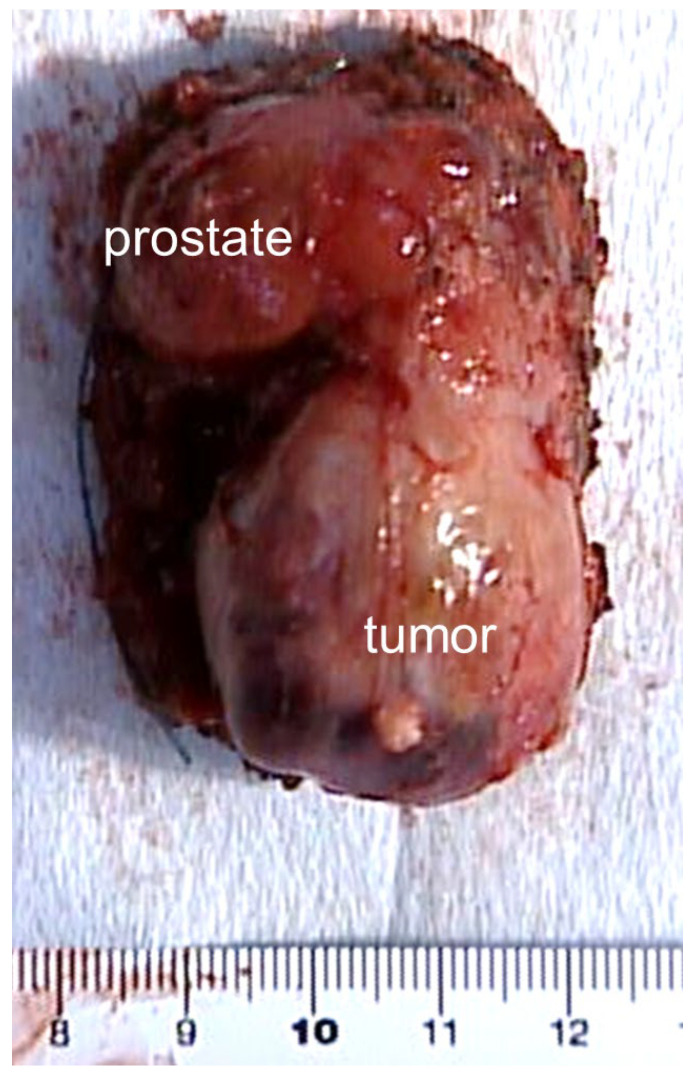
Macroscopic examination reveals a solid tumor measuring 3 × 3.4 cm on the left lobe of the resected prostate.

**Figure 4 medicina-57-01152-f004:**
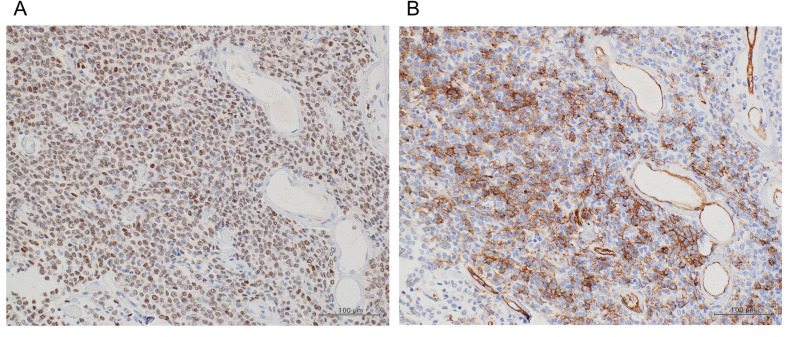
Histopathological examination revealing a well-circumscribed mass composed of spindle cells arranged without any particular architecture (**A**) and associated with variable amounts of collagen and a distinctive vascularization (**B**).

**Figure 5 medicina-57-01152-f005:**
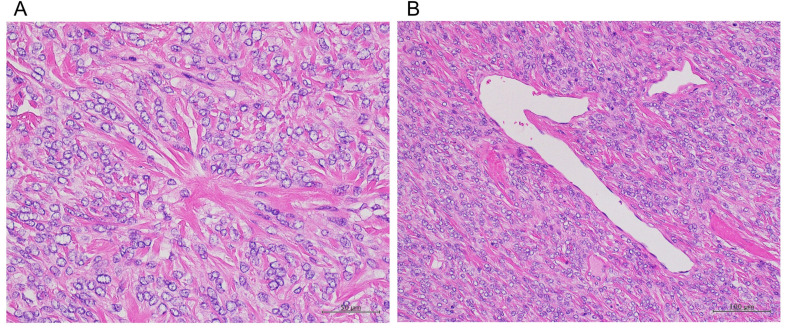
Immunohistochemical analysis of tumor cells showing strong expression of the signal transducer and active tor of transcription 6 (**A**). The tumor cells are partially positive for CD34 (**B**).

**Table 1 medicina-57-01152-t001:** Morphologic features of spindle cell lesions of the prostate [13].

Tumors	Morphology
SFT	(1) Fibroblast-like cells arranged in a disorganized non-repetitive pattern, (2) scattered angulated hemangiopericytic vessels, (3) variable stromal collagenization, (4) occasional cellular and myxoid forms
STUMP	(1) Scattered atypical stromal cells associated with benign glands, (2) resemblance to glandular-stromal hyperplasia yet with hypercellular stroma, (3) extensive myxoid stroma, (4) phyllodes pattern
Stromal sarcoma	(1) Solid growth of overtly malignant cells with storiform, epithelioid, fibrosarcomatous, or patternless pattern, (2) malignant phyllodes-like
Sarcomatoid carcinoma	Mixture of frank epithelial and sarcomatous components such as ‘undifferentiated’ with atypical giant cells or pleomorphic spindle cells
Leiomyoma	Well-organized smooth muscle fascicles lacking mitotic activity
Leiomyosarcoma	Intersecting fascicles showing mitoses, cytologic atypia, and necrosis
Rhabdomyosarcoma	(1) Often spindled, but fusiform or more rounded cells and well-developed rhabdomyoblasts may be present; (2) embryonal subtype consisting of cartilaginous differentiation
IMT	(1) Typical myoblastic features with elongated or ‘stretched-out’ spindle cells, often with a loose myxoid background; (2) nuclei with significant variation in size; (3) nucleoli that may be quite prominent or multiple without nuclear hyperchromasia
GIST	Rectal-based spindle cells that compress and displace the prostate gland

Note: SFT: solitary fibrous tumor, STUMP: stromal tumors of uncertain malignant potential, IMT: inflammatory myofibroblastic tumor, GIST: gastrointestinal stromal tumor. The table is reproduced with permission from Elsevier.

**Table 2 medicina-57-01152-t002:** Immunohistochemical findings of spindle cell lesions of the prostate [13].

	SFT	STUMP	SS	Leiomyo-Sarcoma	Rhabdomyo-Sarcoma	IMT	GIST
CD34	+	+	+	-	-	-	+
SMA	-	±	-	+	+	+	±
Desmin	-	±	-	+	+	+	±
Myogenin	-	-	-	-	+	-	-
c-kit	-	-	-	-	-	±	+
ALK-1	-	-	-	-	-	+	-
PRs	±	+	+	±	-	-	-

Note: SFT: solitary fibrous tumor, STUMP: stromal tumors of uncertain malignant potential, SS: stromal sarcoma, IMT: inflammatory myofibroblastic tumor, GIST: gastrointestinal stromal tumor, SMA: smooth muscle actin, ALK-1: anaplastic lymphoma kinase, PRs: progesterone receptors. The table is reproduced with permission from Elsevier.

## Data Availability

Data are available upon reasonable request.

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
