# Peer review of "Solitary Fibrous Tumor of the Prostate: A Case Report and Literature Review"

_medicina, 2021, doi:10.3390/medicina57111152_

Round 1
Reviewer 1 Report
This article is a rare case report of solitary fibrous tumor (SFT) of the prostate. The diagnosis and treatment process are well documented. It would be better to add a few things to the discussion.
First, please add the differential diagnosis for SFT for example, soft tissue neoplasm and specify the test or differentiation points necessary for differential diagnosis
Second, I would also like the author to add something about the malignant SFT and a point to distinguish it from the benign SFT.
Author Response
6, Oct, 2021
Dr. Miljan Petrovic
Assigned Editor
Medicina
Dear Editor:
Thank you very much for the review of our manuscript titled “Solitary Fibrous Tumor of The Prostate: A Case Report and Literature Review.”
We sincerely appreciate all valuable comments and suggestions, which helped us to improve the quality of our manuscript. Our responses to the Reviewers’ comments are described below in a point-to-point manner. Appropriate changes, suggested by the Reviewers, have been introduced to the manuscript (track-changes mode in the red color font). Let me emphasize our full readiness to make any further improvements to the manuscript.
We hope that our manuscript will be acceptable for publication in the Medicina.
We look forward to hearing from you.
Yours sincerely,
Takuya Koie
Corresponding author
Department of Urology
Gifu University Graduate School of Medicine
1-1 Yanagido, Gifu, Gifu 501-1194, Japan
TEL.: +81-582-30-6338
FAX: +81-582-30-6341
e-mail: goodwin@gifu-u.ac.jp
Responses to the reviewer's comments
We would like to thank the Reviewers for taking the time and effort necessary to review the manuscript. We sincerely appreciate all the valuable comments and suggestions, which helped us to improve the quality of the manuscript.
Response to Reviewer 1
The authors appreciate the reviewer’s comments. The authors’ point-by-point responses to the comments are given below.
- First, please add the differential diagnosis for SFT for example, soft tissue neoplasm and specify the test or differentiation points necessary for differential diagnosis.
Response:
We have added the following sentence, on line 105:
Microscopic and immunohistochemical characteristics of prostatic spindle-cell lesions are shown in Table 1 and 2. [12-14].
We have added the Table 1 and 2.
Table 1 Morphologic features of spindle cell lesions of the prostate.
|
Tumors |
Morphology |
|
SFT |
(1) Fibroblast-like cells arranged in a disorganized non-repetitive pattern, (2) scattered angulated hemangiopericytic vessels, (3) variable stromal collagenization, (4) occasional cellular and myxoid forms |
|
STUMP |
(1) Scattered atypical stromal cells associated with benign glands, (2) resemblance to glandular-stromal hyperplasia yet with hypercellular stroma, (3) extensive myxoid stroma, (4) phyllodes pattern |
|
Stromal sarcoma |
(1) Solid growth of overtly malignant cells with storiform, epithelioid, fibrosarcomatous, or patternless pattern, (2) malignant phyllodes-like |
|
Sarcomatoid carcinoma |
Mixture of frank epithelial and sarcomatous components such as ‘undifferentiated’ with atypical giant cells or pleomorphic spindle cells |
|
Leiomyoma |
Well-organized smooth muscle fascicles lacking mitotic activity |
|
Leiomyosarcoma |
Intersecting fascicles showing mitoses, cytologic atypia, and necrosis |
|
Rhabdomyosarcoma |
(1) Often spindled, but fusiform or more rounded cells and well-developed rhabdomyoblasts may be present; (2) embryonal subtype consisting of cartilaginous differentiation |
|
IMT |
(1) Typical myoblastic features with elongated or ‘stretched-out’ spindle cells, often with a loose myxoid background; (2) nuclei with significant variation in size; (3) nucleoli that may be quite prominent or multiple without nuclear hyperchromasia |
|
GIST |
Rectal-based spindle cells that compress and displace the prostate gland |
Note: SFT: solitary fibrous tumor, STUMP: stromal tumors of uncertain malignant potential, IMT: inflammatory myofibroblastic tumor, GIST: gastrointestinal stromal tumor
Table 2 Immunohistochemical findings of spindle cell lesions of the prostate
|
|
SFT |
STUMP |
SS |
Leiomyo-sarcoma |
Rhabdomyo-sarcoma |
IMT |
GIST |
|
CD34 |
+ |
+ |
+ |
- |
- |
- |
+ |
|
SMA |
- |
± |
- |
+ |
+ |
+ |
± |
|
Desmin |
- |
± |
- |
+ |
+ |
+ |
± |
|
Myogenin |
- |
- |
- |
- |
+ |
- |
- |
|
c-kit |
- |
- |
- |
- |
- |
± |
+ |
|
ALK-1 |
- |
- |
- |
- |
- |
+ |
- |
|
PRs |
± |
+ |
+ |
± |
- |
- |
- |
Note: SFT: solitary fibrous tumor, STUMP: stromal tumors of uncertain malignant potential, SS: stromal sarcoma, IMT: inflammatory myofibroblastic tumor, GIST: gastrointestinal stromal tumor, SMA: smooth muscle actin, ALK-1: anaplastic lymphoma kinase, PRs: progesterone receptors
We have added the following reference:
- Galosi, A.B.; Mazzucchelli, R.; Scarpelli, M.; Lopez-Beltran, A.; Cheng, L.; Muzzonigro, G.; Montironi, R. Solitary fibrous tumour of the prostate identified on needle biopsy. Eur Urol 2009, 56, 564-567.
- Second, I would also like the author to add something about the malignant SFT and a point to distinguish it from the benign SFT.
Response:
We have added the following sentences on line 128:
Recently, risk stratification models have identified malignant SFT, which behaves aggressively [19].
We have added the following sentences on line 131:
Demicco et al. [20] developed a risk stratification model based on patients’ age, tumor size, and mitotic figures. Patients older than 55 years with tumor size greater than 15 cm, and mitotic counts greater than 4 mitoses/10 HPFs had the highest risk of metastases or tumor death. In addition, telomerase reverse transcriptase gene promoter mutations were strongly associated with higher age, larger tumor size, and higher risk classification [22].
We have added the following reference:
- Olson, N.J.; Linos, K. Dedifferentiated Solitary Fibrous Tumor: A Concise Review. Arch Pathol Lab Med 2018, 142, 761-766.
- Demicco, E.G.; Park, M.S.; Araujo, D.M.; Fox, P.S.; Bassett, R.L.; Pollock, R.E.; Lazar, A.J.; Wang, W.L. Solitary fibrous tumor: a clinicopathological study of 110 cases and proposed risk assessment model. Mod Pathol 2012, 25, 1298-1306.
- Bahrami, A.; Lee, S.; Schaefer, I.M.; Boland, J.M.; Patton, K.T.; Pounds, S.; Fletcher, C.D. TERT promoter mutations and prognosis in solitary fibrous tumor. Mod Pathol 2016, 29, 1511-1522.
Reviewer 2 Report
Dear Editor,
thank you for giving me the opportunity to revise this paper submitted to Medicina'. It is a clinical report which addresses a case of a solitary fibrous tumor (SFT) of the prostate. As the authors stated in the introduction, this type of lesion very rarely involves the prostate. STF is a soft tissue lesion (mesenchymal origin), mostly due to a paracentric inversion involving chromosome 12q, resulting in NAB2-STAT6 gene fusion (Nat Genet 2013;45:131, Nat Genet 2013;45:180). Although most SFTs are benign, many studies have tried to identify histologic features that can be useful to predict which tumors will behave in an aggressive manner (Olson NJ, Linos K. Dedifferentiated Solitary Fibrous Tumor: A Concise Review. Arch Pathol Lab Med. 2018 Jun;142(6):761-766. doi: 10.5858/arpa.2016-0570-RS. PMID: 29848035.).
Here, Yasumichi Takeuchi et al. presented a very interesting case report. I hope they will address my concern for improving their work. In 2017, the corresponding author reported a case of SFT of the prostate which was initially misdiagnosed as prostate cancer (see Osamu et al reference below).
Minor concerns
Abstract. Line 14. There is a concern about the patient's age. Here you state that it is 43, but in the manuscript, it becomes 42. Please clarify.
Please expand the introduction. Since a case report should be didactic, you should expand the introduction. For instance, prostate SFT is a diagnostic challenge (see Nishith N, Gupta M, Kaushik N, Sen R. Solitary Fibrous Tumor of the Prostate: A Diagnostic Challenge: A Case Report. Iran J Pathol. 2020;15(1):41-44. doi:10.30699/IJP.2019.104669.2069), especially for pathologists. According to Osamu et al (Osamu S, Murasawa H, Imai A, Hatakeyama S, Yoneyama T, Hashimoto Y, Koie T, Ohyama C. Solitary Fibrous Tumor of the Prostate Which Was Initially Misdiagnosed as Prostate Cancer. Case Rep Urol. 2017; 2017():3594914) it can simulate poorly differentiated adenocarcinoma and various mesenchymal neoplasms of the prostate.
Line 48. Insert a comma after the word 'intensity'.
Since you report that 'there was firm adhesion between the tumor and rectal' (Line 61 and 62), more details about the RALP approach must be offered. What about nerve-sparing? The RALP-associated neurologic injuries may occur even when performed by highly experienced surgeons. Pudendal nerve? obturator nerve? (Neuropathic painful complications due to endopelvic nerve lesions after robot-assisted laparoscopic prostatectomy: Three case reports. Medicine (Baltimore). 2019;98(46):e18011. doi:10.1097/MD.0000000000018011).
This paper can be interesting for different professionals including urologists, oncologists, pathologists, and others. Consider that, according to McKenney [reference number 7], benign sub-epithelial connective tissue (or benign prostatic hyperplasia) can simulate SFT. Thus, I'd like if the pathological finding and the differential diagnosis will insert in a dedicated table. For example, you should indicate the dufferential diagnosis with spindle lesions (e.g., Stromal tumors of uncertain malignant potential, Stromal sarcoma, Sarcomatoid carcinoma, Leiomyoma, and others) (you can see A. B. Galosi, R. Mazzucchelli, M. Scarpelli et al., “Solitary fibrous tumour of the prostate identified on needle biopsy,” European Urology, vol. 56, no. 3, pp. 564–567, 2009. ).
Line 101. Put reference number after 'et al.'
Figures. You presented very interesting pictures. Have you a picture of the specimen (namely the macroscopic finding of the resected prostate)?
Finally, several references must be necessarily added (see all the comments).
Author Response
6, Oct, 2021
Dr. Miljan Petrovic
Assigned Editor
Medicina
Dear Editor:
Thank you very much for the review of our manuscript titled “Solitary Fibrous Tumor of The Prostate: A Case Report and Literature Review.”
We sincerely appreciate all valuable comments and suggestions, which helped us to improve the quality of our manuscript. Our responses to the Reviewers’ comments are described below in a point-to-point manner. Appropriate changes, suggested by the Reviewers, have been introduced to the manuscript (track-changes mode in the red color font). Let me emphasize our full readiness to make any further improvements to the manuscript.
We hope that our manuscript will be acceptable for publication in the Medicina.
We look forward to hearing from you.
Yours sincerely,
Takuya Koie
Corresponding author
Department of Urology
Gifu University Graduate School of Medicine
1-1 Yanagido, Gifu, Gifu 501-1194, Japan
TEL.: +81-582-30-6338
FAX: +81-582-30-6341
e-mail: goodwin@gifu-u.ac.jp
Responses to the reviewer's comments
We would like to thank the Reviewers for taking the time and effort necessary to review the manuscript. We sincerely appreciate all the valuable comments and suggestions, which helped us to improve the quality of the manuscript.
Response to Reviewer 2
The authors appreciate the reviewer’s comments. The authors’ point-by-point responses to the comments are given below.
Thank you for giving me the opportunity to revise this paper submitted to Medicina'. It is a clinical report which addresses a case of a solitary fibrous tumor (SFT) of the prostate. As the authors stated in the introduction, this type of lesion very rarely involves the prostate. STF is a soft tissue lesion (mesenchymal origin), mostly due to a paracentric inversion involving chromosome 12q, resulting in NAB2-STAT6 gene fusion (Nat Genet 2013;45:131, Nat Genet 2013;45:180). Although most SFTs are benign, many studies have tried to identify histologic features that can be useful to predict which tumors will behave in an aggressive manner (Olson NJ, Linos K. Dedifferentiated Solitary Fibrous Tumor: A Concise Review. Arch Pathol Lab Med. 2018 Jun;142(6):761-766. doi: 10.5858/arpa.2016-0570-RS. PMID: 29848035.).
Here, Yasumichi Takeuchi et al. presented a very interesting case report. I hope they will address my concern for improving their work. In 2017, the corresponding author reported a case of SFT of the prostate which was initially misdiagnosed as prostate cancer (see Osamu et al reference below).
Response:
We have revised the text in the following sections according to the reviewer’s recommendations.
- Abstract. Line 14. There is a concern about the patient's age. Here you state that it is 43, but in the manuscript, it becomes 42. Please clarify. Response:
The authors have revised this point according to the reviewer’s recommendation.
- Please expand the introduction. Since a case report should be didactic, you should expand the introduction. For instance, prostate SFT is a diagnostic challenge (see Nishith N, Gupta M, Kaushik N, Sen R. Solitary Fibrous Tumor of the Prostate: A Diagnostic Challenge: A Case Report. Iran J Pathol. 2020;15(1):41-44. doi:10.30699/IJP.2019.104669.2069), especially for pathologists. According to Osamu et al (Osamu S, Murasawa H, Imai A, Hatakeyama S, Yoneyama T, Hashimoto Y, Koie T, Ohyama C. Solitary Fibrous Tumor of the Prostate Which Was Initially Misdiagnosed as Prostate Cancer. Case Rep Urol. 2017; 2017():3594914) it can simulate poorly differentiated adenocarcinoma and various mesenchymal neoplasms of the prostate.
Response:
We have added the following sentence on line 34:
STF, a soft tissue lesion of mesenchymal origin, has been recently determined to be associated with a paracentric inversion involving chromosome 12q that results in a NAB2-STAT6 fusion gene [4,5].
We have added the following sentence on line 38:
However, the diagnosis of mesenchymal neoplasms of the prostate is often difficult for pathologists because of their overlapping histomorphological and immunohistochemical (IHC) characteristics [7]. In fact, we previously encountered a patient with prostate SFT that was initially misdiagnosed as prostate cancer, based on histopathological examination of prostate biopsy specimen [8].
- Line 48. Insert a comma after the word 'intensity'.
Response:
The authors have revised this point according to the reviewer’s recommendation.
- Since you report that 'there was firm adhesion between the tumor and rectal' (Line 61 and 62), more details about the RALP approach must be offered. What about nerve-sparing? The RALP-associated neurologic injuries may occur even when performed by highly experienced surgeons. Pudendal nerve? obturator nerve? (Neuropathic painful complications due to endopelvic nerve lesions after robot-assisted laparoscopic prostatectomy: Three case reports. Medicine (Baltimore). 2019;98(46):e18011. c.
Response:
We have added the following sentence on line 69:
In this case, we could spare the neurovascular bundle on the right side. There have been no surgery-related complications, including neurologic complications [11].
- This paper can be interesting for different professionals including urologists, oncologists, pathologists, and others. Consider that, according to McKenney [reference number 7], benign sub-epithelial connective tissue (or benign prostatic hyperplasia) can simulate SFT. Thus, I'd like if the pathological finding and the differential diagnosis will insert in a dedicated table. For example, you should indicate the differential diagnosis with spindle lesions (e.g., Stromal tumors of uncertain malignant potential, Stromal sarcoma, Sarcomatoid carcinoma, Leiomyoma, and others) (you can see A. B. Galosi, R. Mazzucchelli, M. Scarpelli et al., “Solitary fibrous tumour of the prostate identified on needle biopsy,” European Urology, vol. 56, no. 3, pp. 564–567, 2009.).
Response:
We have added the Table 1 and 2.
Table 1 Morphologic features of spindle cell lesions of the prostate.
|
Tumors |
Morphology |
|
SFT |
(1) Fibroblast-like cells arranged in a disorganized non-repetitive pattern, (2) scattered angulated hemangiopericytic vessels, (3) variable stromal collagenization, (4) occasional cellular and myxoid forms |
|
STUMP |
(1) Scattered atypical stromal cells associated with benign glands, (2) resemblance to glandular-stromal hyperplasia yet with hypercellular stroma, (3) extensive myxoid stroma, (4) phyllodes pattern |
|
Stromal sarcoma |
(1) Solid growth of overtly malignant cells with storiform, epithelioid, fibrosarcomatous, or patternless pattern, (2) malignant phyllodes-like |
|
Sarcomatoid carcinoma |
Mixture of frank epithelial and sarcomatous components such as ‘undifferentiated’ with atypical giant cells or pleomorphic spindle cells |
|
Leiomyoma |
Well-organized smooth muscle fascicles lacking mitotic activity |
|
Leiomyosarcoma |
Intersecting fascicles showing mitoses, cytologic atypia, and necrosis |
|
Rhabdomyosarcoma |
(1) Often spindled, but fusiform or more rounded cells and well-developed rhabdomyoblasts may be present; (2) embryonal subtype consisting of cartilaginous differentiation |
|
IMT |
(1) Typical myoblastic features with elongated or ‘stretched-out’ spindle cells, often with a loose myxoid background; (2) nuclei with significant variation in size; (3) nucleoli that may be quite prominent or multiple without nuclear hyperchromasia |
|
GIST |
Rectal-based spindle cells that compress and displace the prostate gland |
Note: SFT: solitary fibrous tumor, STUMP: stromal tumors of uncertain malignant potential, IMT: inflammatory myofibroblastic tumor, GIST: gastrointestinal stromal tumor
Table 2 Immunohistochemical findings of spindle cell lesions of the prostate
|
|
SFT |
STUMP |
SS |
Leiomyo-sarcoma |
Rhabdomyo-sarcoma |
IMT |
GIST |
|
CD34 |
+ |
+ |
+ |
- |
- |
- |
+ |
|
SMA |
- |
± |
- |
+ |
+ |
+ |
± |
|
Desmin |
- |
± |
- |
+ |
+ |
+ |
± |
|
Myogenin |
- |
- |
- |
- |
+ |
- |
- |
|
c-kit |
- |
- |
- |
- |
- |
± |
+ |
|
ALK-1 |
- |
- |
- |
- |
- |
+ |
- |
|
PRs |
± |
+ |
+ |
± |
- |
- |
- |
Note: SFT: solitary fibrous tumor, STUMP: stromal tumors of uncertain malignant potential, SS: stromal sarcoma, IMT: inflammatory myofibroblastic tumor, GIST: gastrointestinal stromal tumor, SMA: smooth muscle actin, ALK-1: anaplastic lymphoma kinase, PRs: progesterone receptors
- Line 101. Put reference number after 'et al.'
Response:
The authors have revised this point according to the reviewer’s recommendation.
- Figures. You presented very interesting pictures. Have you a picture of the specimen (namely the macroscopic finding of the resected prostate)?
Response:
The authors have added the following sentence on line 74:
Macroscopic examination revealed a solid, yellowish-white tumor measuring 3 × 3.4 cm on the left side of the resected prostate (Figure 3).
The authors have added the picture of the specimen in Figure 3.
Figure 3. Macroscopic examination reveals a solid tumor measuring 3 × 3.4 cm on the left lobe of the resected prostate.
- Finally, several references must be necessarily added (see all the comments).
Response:
We have added the following reference:
- Chmielecki, J.; Crago, A.M.; Rosenberg, M.; O'Connor, R.; Walker, S.R.; Ambrogio, L.; Auclair, D.; McKenna, A.; Heinrich, M.C.; Frank, D.A.; Meyerson, M. Whole-exome sequencing identifies a recurrent NAB2-STAT6 fusion in solitary fibrous tumors. Nat Genet 2013, 45, 131-132.
- Robinson, D.R.; Wu, Y.M.; Kalyana-Sundaram, S.; Cao, X.; Lonigro, R.J.; Sung, Y.S.; Chen, C.L.; Zhang, L.; Wang, R.; Su, F.; Iyer, M.K.; Roychowdhury, S.; Siddiqui, J.; Pienta, K.J.; Kunju, L.P.; Talpaz, M.; Mosquera, J.M.; Singer, S.; Schuetze, S.M.; Antonescu, C.R.; Chinnaiyan, A.M. Identification of recurrent NAB2-STAT6 gene fusions in solitary fibrous tumor by integrative sequencing. Nat Genet 2013, 45, 180-185.
- Nishith, N.; Gupta, M.; Kaushik, N.; Sen R. Solitary Fibrous Tumor of the Prostate: A Diagnostic Challenge: A Case Report. Iran J Pathol 2020, 15, 41-44.
- Osamu S, Murasawa H, Imai A, Hatakeyama S, Yoneyama T, Hashimoto Y, Koie T, Ohyama C. Solitary Fibrous Tumor of the Prostate Which Was Initially Misdiagnosed as Prostate Cancer. Case Rep Urol 2017, 2017, 3594914.
- Cascella, M.; Quarto, G.; Grimaldi, G.; Izzo, A.; Muscariello, R.; Castaldo, L.; Di Caprio, B.; Bimonte, S.; Del Prete, P.; Cuomo, A.; Perdonà, S. Neuropathic painful complications due to endopelvic nerve lesions after robot-assisted laparo-scopic prostatectomy: Three case reports. Medicine (Baltimore) 2019, 98, e18011.
- Galosi, A.B.; Mazzucchelli, R.; Scarpelli, M.; Lopez-Beltran, A.; Cheng, L.; Muzzonigro, G.; Montironi, R. Solitary fibrous tumour of the prostate identified on needle biopsy. Eur Urol 2009, 56, 564-567.
- Olson, N.J.; Linos, K. Dedifferentiated Solitary Fibrous Tumor: A Concise Review. Arch Pathol Lab Med 2018, 142, 761-766.
- Demicco, E.G.; Park, M.S.; Araujo, D.M.; Fox, P.S.; Bassett, R.L.; Pollock, R.E.; Lazar, A.J.; Wang, W.L. Solitary fibrous tumor: a clinicopathological study of 110 cases and proposed risk assessment model. Mod Pathol 2012, 25, 1298-1306.
- Bahrami, A.; Lee, S.; Schaefer, I.M.; Boland, J.M.; Patton, K.T.; Pounds, S.; Fletcher, C.D. TERT promoter mutations and prognosis in solitary fibrous tumor. Mod Pathol 2016, 29, 1511-1522.